# Strategic Decisions for Sustainable Management at Significant Tourist Sites

**Robert M. Mackay \*, Roberto Minunno and Gregory M. Morrison** 

Curtin University Sustainability Policy (CUSP) Institute, Curtin University, Bentley 6102, Australia;
Roberto.Minunno@curtin.edu.au (R.M.); Greg.Morrison@curtin.edu.au (G.M.M.)
\* Correspondence: Robert.Mackay@postgrad.curtin.edu.au; Tel.: +61-409-302-595

**Abstract:** This research explores how tourist site management and human attitudes and behaviours can help decrease the pressure of tourism on the environment. Estimates show that, together with ancillary sectors, the tourism industry is expected to contribute approximately 6.5 gigatons of greenhouse gases by 2025. These emissions are primarily a result of tourists favouring air travel and luxury experiences that require more energy outputs. Additionally, tourism continues to grow and has become a routine activity for the middle class who travel more regularly on an annual basis. With growing middle classes in many developing countries, the number of tourists who will be able to afford recreational travel is estimated to increase exponentially. The pressures and demands of increasing tourist numbers can strain vulnerable natural sites. These predictions show that changes within the tourism industry fabric are necessary. Against this backdrop, this research employs a combined methodology. A survey methodology was employed to explore tourist attitudes towards tourism sites and their behaviours and decision making with a top-down and bottom-up approach. Additionally, an interview methodology of tourism field experts was employed to investigate the attitudes of the industry and how consumer behaviours may be influenced. Findings from the survey and interview discussions were employed to inform four managerial aspects. First, the ticket price of the tourist experience should be proportional to the value proposition of the experience. Second, a government-led framework could guide businesses towards sustainable management and educate their tourists on greener practices. Third, businesses could integrate sustainability issues into their marketing and advertising to create awareness and ensure the longevity of the site. Lastly, tourism bodies and businesses could increase their partnerships with local custodians to add cultural value and understand the visitor experience.

**Keywords:** sustainable tourism; last chance tourism; management strategies; tourist sites

## 1. Introduction

The tourism industry has historically been developed for people seeking a hedonistic experience, an escape from the normality of everyday life, and tourists often leave their values at home, including those related to environmental sustainability. As tourism becomes more accessible and affordable to most socio-economic levels, the number of people travelling across the globe continues to grow and, prior to the COVID-19 pandemic, was expected to reach 1.8 billion by 2030 [1]. Whilst this continuously growing industry has proven beneficial in creating jobs and increasing the wealth of many localities, the adverse environmental impacts of tourism are also steadily increasing. Among these impacts, an estimate conducted on 160 countries showed that the tourism industry contributes approximately 4.5 gigatons of greenhouse gases per annum [2]. That figure is significantly higher than previous estimates, as it combines the ancillary sectors of the industry, and it is predicted to climb to 6.5 gigatons of greenhouse gases by 2025 [2]. Further, tourist experiences consume a notable amount of energy and

thus create additional and significant carbon dioxide emissions [3]. Many of these emissions are related to electricity usage and, more prevalently, air and road transport [4]. Interestingly, tourist activities were recorded as more energy intensive in comparison to tourist attractions, and a combined countries' focus on tourist activities would lead to greater demand for energy use [3].

A further contribution that increases the adverse environmental impact of tourism is the phenomenon of overtourism [5]. Overtourism is typically considered antithetical to the ideal of responsible tourism [6]. Overtourism is defined by a situation where the number of tourists impairs the quality of life for communities and quality of experience for other tourists. An example could be seen in Boracay Island, Philippines, where the large number of tourists have strained the natural environment and inadequate facilities such as water management and sewerage [7]. Whilst the impacts of overtourism are not exclusive to the natural environment (as seen, for example, through the damage caused to the ecology, economics, infrastructure, and culture in Venice, Italy), that is where their impacts will have the most enduring effects [8].

It is important to note that tourism has contributed to an increase in economic sustainability in local communities, and social sustainability has improved through awareness of local cultures and a platform for small business owners to make a living. However, on the other hand, these positive changes cannot be dismissed in favour of environmental preservation. A commonly discussed theme is how the changing climate is impacting tourism in turn by altering seasons, destination popularity, and thus revenues [9]. Extreme and unpredictable weather is reducing tourist traffic; for example, hurricanes severely affect the Caribbean islands during increasingly severe hurricane seasons, and in turn, the islands have experienced a significant decrease in tourism traffic [10]. Moreover, changes in temperature are shifting tourist seasons altogether (October in Botswana is no longer a popular travel period due to significant increases in temperature and erratic precipitation [11]), and some rain seasons prevent river inundations, pausing all river activities [12]. Unpredictability of climate and weather will also come at a cost as insurance and travel companies consider claims and guarantees on tourist experiences, many of which are weather-dependent [10].

There is often debate about what sustainability means and in which context it should be understood. Through the lens of this paper, sustainability can be viewed in two ways. The primary focus of sustainability here in tourism, and specifically for tourist sites, is that for the industry to be considered sustainable, it should leave sites in equal or improved condition to when they were found or established (similarly to the definition that [13] applied in the context of bioeconomy). The second lens to view sustainable tourism through is the idea of balancing environmental, economic, and social sustainability [14].

The literature suggests that, as of the 1990s, the idea of sustainable development had already received widespread recognition [15]. This challenges whether the idea of sustainability has gained traction due to a genuine societal understanding or as a popular trend, suggesting that action is more tokenistic than driven by change [15]. Similarly, the literature associates the idea of ecotourism as a concept that values principles of sustainability, respect for and appreciation of the environment, and respect and understanding of a region or environment's indigenous context and background [16]. However, in the tourism sector, environmental, economic, and social sustainability have not yet been harmonized. Problematically, revenue often takes precedence over conservation at tourist sites, and using the Galapagos, Ecuador, as an example, there is a concern that a focus on conservation will be to the detriment of economic growth [17]. More than half of World Heritage sites lack environmental sustainability management integrated into their business plans, leaving a gap in ensuring the longevity of sites for future generations to enjoy and to sustain reliant businesses [18]. Interestingly, at the same time, awareness is increasing between managers in the tourism industry regarding the notion that their sustained success is dependent upon the environment that they operate in. Conservation efforts are in the interest of the business itself: if the site is endangered, the revenue ceases [19].

Although many studies have proposed management plans for specific tourism sites (e.g., South Sulawesi Province in Indonesia [20], Porto Santo in Madeira, Portugal [21], Polish National Parks [22]),

research that incorporates tourists' behavior and opinions is scarce, especially in the context of sustainability and overtourism [21]. To address this knowledge gap, the main objective of this paper is to propose several strategies, which managers could apply to their tourist sites to decrease the environmental pressure of tourists, without compromising the sites income.

To inform the proposed strategies, both quantitative studies into the tourism consumer and qualitative studies with tourism industry experts have been conducted. Therefore, a combined methodology was employed in this paper. It constitutes a survey methodology to gather tourists' behaviours towards environmental sustainability, and an interview methodology applied to tourism site managers to explore, through a top-down approach, managers' drivers and barriers towards the adoption of sustainable operations.

Results show aligning perspectives on how the industry may shift towards a greater sustainability balance. More specifically, the survey conducted on 165 tourists found an overarching willingness for individuals to alter their behaviours and decision making if provided with more information about sustainability. There is, however, a variance between individuals translating their intentions and attitudes pre/post vacation to actual behaviours whilst visiting. As part of this research, interviewed experts collectively produced a narrative whereby they agreed that human behaviours were at the crux of sustainability issues at tourist sites, but that industry bodies and businesses could do more to include the tourist in the sustainability discussion and make it easier for tourists to make responsible decisions.

As sustainability becomes a more socialised and marketable concept, there is a reflective willingness on the consumers behalf to become more responsible in their behaviours and decision making. To ensure this goodwill translates to actual behaviours at tourist sites, partnerships between industry and tourists are paramount. Updated strategies will likely need to be employed by the tourism industry to alter timeless and systemic behaviours. Managers could operate towards changing the costs of experiences to more closely align to sites' values, a framework could be developed that guides businesses in operating sustainably, and including sustainability values into the experience and establishing partnerships with local custodians could assist in shifting how people engage with the tourism industry and the tourist experience.

This paper unfolds in five chapters. Section 2 explains the details of the survey and interview methods adopted to carry out the research. Section 3 proposes the answers that had been collected from both surveys and interviews. Section 4 shows the managerial strategies that had been distilled from the surveys and interviews. Section 5 summarises the main contributions and limitations of this study.

*Theoretical Background*

Early discussions of sustainable tourism posited the theory as a "normative orientation" to redirect societal behaviours and systems towards sustained development [23]. Concepts of "overtourism", "last chance tourism", and "ecotourism" have only added congestion to the conversation, often making sustainability efforts misdirected or seem futile.

Weaver [24] describes sustainable tourism as one of two things—the continued, sustained growth or intensification of tourism, or the pursuit of alternate tourism habits and a halt to mass tourism. Noting that the understanding of the principle changes according to the consensus surrounding it, this means it often has very little practicality [24]. In simple terms, sustainable tourism is considered to be about maximising the positive benefits of tourism and reducing those that would be considered detrimental [25].

"Overtourism" is defined by when the number of tourists impairs the quality of life for communities and quality of tourist experience and is typically antithetical to sustainable tourism [6]. This idea of a "growth paradox", where increasing tourist numbers overcrowds sites, stresses the infrastructure, and damages the local community [25], is particularly evident in the South Pacific, where the increase in tourism has exacerbated societal inequalities within communities, placed a disproportionate demand on the regions infrastructure, and forced Pacific nations to rely on foreign capital [26].

Debate around the purpose of sustainability within the scope of tourism has created a misunderstanding on behalf of the consumer and manager on how to approach sustainability, and thus efforts toward protection and conservation have been fragmented or misdirected. Critical to popular thought is the idea of integrating the need for mass awareness, socialisation, and education of sustainability concepts within the public to maximise the influence of travellers (Kummerer, 2018). Multiple authors [14,23,27] conclude that if a sustainable tourism model is to be successful, institutional support including divisional policy and strategic management are critical.

One particular theoretical observation was the shift in framing of the issue towards changing social behaviours and the choices of individual actors as a crucial element of sustainability efforts. Quantitative data collected by National Geographic indicate that whilst there is growing support for the idea of sustainable tourism, few of those supporters actually understood the concept or what a commitment to sustainable tourism looked like [5]. Collecting data from consumers is crucial to understanding the mass attitudes to sustainability, and thus this informs the study [28].

Assessed theories suggest the sustainable tourism field of study is maturing and becoming more multidisciplinary. Commentary is, however, often paid to how the existing structures and frameworks are failing, with few sustainability or policy initiatives recommended to offset these failings. Prominently, the literature shows that methods including interviews with experts and other observational methods offer valuable insights into how to manage the future sustainability of the industry [29]. Further sustainability themes often indicate the integration of human behavior in management is critical [30,31]. This observation has informed the methodology and focus of this paper.

## 2. Materials and Methods

Managerial strategies can be applied in tourism sites to limit the adverse environmental impact of phenomena such as last-minute tourism [32] and overtourism [33]. Problematically, such strategies must not only make sure that the site is managed in a way that decreases the pressure of tourists on the environment, but also without harming the economic sustainability of such sites. These strategies need to also provide an adequate level of customer satisfaction upon their visit to the site, lest they reduce the intended purpose of the site altogether. To keep these aspects in consideration, this paper explores managers' and tourists' opinions on environmental, economic, and social sustainability. Therefore, the theoretical framework applied in this research combines both bottom-up and top-down theories (Figure 1). The bottom-up theory is based on the idea that individuals live in a social context that combines with their general perception of phenomena, such as sustainable tourism, and therefore shapes their ideas about the world around them [34]. By gathering their collective ideas through extended surveys, it is possible to distill a common thought in a specific context, which is paramount for fostering innovation in the management sector [35]. Further, interviewing managers and field experts in tourism can be seen as a top-down approach of qualitative research, which in the case of this research was adopted to explore managers' perspective in depth [36]. Therefore, interviews can foster an understanding of primary management strategies by interviewing managers and field experts [37].

The combination of bottom-up and top-down approaches is useful in research, as from the bottom-up approach it is possible to understand the community knowledge that is necessary to support managerial strategies, whilst a top-down approach is fundamental to balance the customer perspective with experts' opinions [38].

Therefore, this paper has explored, through a top-down and bottom-up approach, the behaviours and decision making of participants in the tourism industry, including consumers of tourism (tourists) and the businesses that supply the service. Human behaviour is believed to be responsible for great impact on the tourism industry and tourist sites of significance in particular. More specifically, the volume of visitors at singular times strains the site (e.g., Milford Sound where concentrated tourist numbers significantly impact dolphin residency patterns in the fiord and create travel congestion on the roads [39]). This issue is exacerbated by visitors not understanding their role in the environment

through a lack of attachment to the place [40]. Whilst human tourist behaviours have not changed over the course of tourism history, the volume of tourists has widely increased [41].

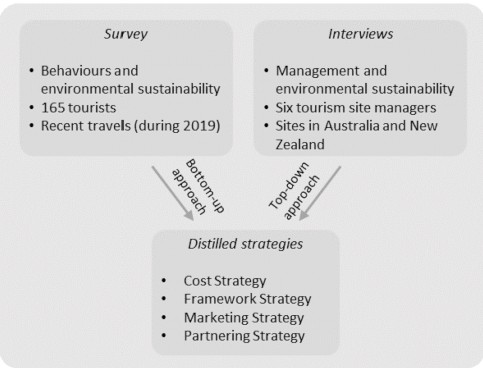

**Figure 1.** Research theoretical framework, including methods employed and distilled strategies.

To assist in understanding the industry, interviews with experts and representatives of the field were organised. By comparing the attitudes and behaviours of individuals at tourist sites with the observations and understandings of the industry, it was possible to study how to influence tourists towards more responsible behaviours. Therefore, the study employed a combination of a survey (quantitative) method to gather data on the way tourists behave and an interview (qualitative) method to explore the approach that tourist sites managers have towards sites' environmental and economic sustainability. Combining research methods provided a range of quantitative and qualitative data that allowed for assessments against the attitudes and behaviours of tourists to be made. The idea of mixed methods is increasingly developing in social science methodologies, as it allows for theory building to be analysed and measured through the quantitative metrics [36,42]. By integrating approaches and findings from both qualitative and quantitative fields, an inclusive account of the discovered findings can be developed in a way that proposes future pragmatic action [43].

The survey format gathered quantitative data in the form of eight closed questions (Table 1) to target specific responses and force the participant to choose. By posing yes or no questions, the surveys provided an insight into the consumers' motivations and decision-making process and also allowed for easier measurement and analysis of the results against qualitative data [44]. Surveys are useful in understanding social and psychological patterns, and thus it was designed to assess whether human behaviours and choices were currently informed, and what the outcome would be if the respondents were provided with more information concerning the tourist site they visited [45]. Therefore, the survey was constructed to target a wide range of tourists and asked the questions in an order that isolated the human thought process (i.e., behaviours to decision making). The survey was run through Qualtrics.com, an online survey platform, and was advertised through social media websites. The survey was completed by 165 participants. Participants in the survey were from all demographics and locations with the only clause being that they must have travelled during the year before the survey, i.e., in 2019. Although initially most respondents were born after 1980, additional age groups were targeted through social media networks to ensure a scope of demographics was achieved.

**Table 1.** List of survey yes/no questions.

| Survey Questions |
| --- |
| 1. Do you consider the sustainability of tourist sites prior to visiting? |
| 2. Was your decision to visit tourist sites in the last year informed by the environmental, cultural, or historical significance of the site? |
| 3. Would you adjust your behaviours if you had more information about the sustainability of the site? |
| 4. Do you think about why a site becomes a tourist destination? |
| 5. Do you consider your impact on the local environment and economy as a tourist? |
| 6. Would you be more inclined to visit a tourist site if you knew it was going to disappear in the coming years? |
| 7. Do you think your decisions as a tourism consumer would change if you knew more about sustainability? |
| 8. Do you think tourist sites should have sustainability management plans in place? |

For the interview portion of the research, participants were asked seven open questions (Table 2) that encouraged a discourse around the key issues within the tourism industry and how behaviours may be altered [46]. Participants were selected based on their proximity to the tourism industry with field experts chosen from both Australia and New Zealand to draw a comparison between approaches. Each interview was conducted for approximately 30 min and recorded to accomplish verbatim referencing. Interview questions were designed to gauge the opinions of experts on what they personally thought were the biggest issues in the tourism industry including attitudes, approaches, and opportunities. Interview participants were asked to respond according to how the tourism industry appeared prior to the global COVID-19 pandemic (prior to March 2020) to effectively measure their responses.

**Table 2.** List of open-ended interview guiding questions.

| Interview Questions |
| --- |
| 1. What do you think are the biggest issues facing the tourism industry in regard to sustainability? |
| 2. Do you think tourism could be considered a sustainable industry? |
| 3. Why do you think many tourist sites do not have sustainability management plans in place? |
| 4. Do you think sustainability is approached differently in regard to the type of tourist site? i.e., cultural, environmental, historical. |
| 5. What balance should be struck between managing the sustainability of the tourist site versus the sustainability of adjacent issues such as local economies and cultural integrity? |
| 6. What could be done to influence the tourism industries' response to sustainability? |
| 7. How much weight do human behaviours carry in regard to sustainability and how can we best influence this? |

## 3. Results

### 3.1. Survey Results Analysis

As introduced in Section 1, in the context of the survey, sustainability is used as a term to define leaving a site in the same or better condition after visiting, which protects or enhances the experience for future generations [47]. Therefore, the premise of this study is to explore ways to improve endurance and longevity of tourist sites. A high-level capture of the survey results can be found in Table 3.

**Table 3.** Survey results percentage. A total of 165 people were interviewed.

| Survey Questions (165 Surveyed People) | Yes (%) | No (%) |
|---|---|---|
| 1. Do you consider the sustainability of tourist sites prior to visiting? | 47.9 | 52.1 |
| 2. Was your decision to visit tourist sites in the last year informed by the environmental, cultural, or historical significance of the site? | 69.5 | 30.5 |
| 3. Would you adjust your behaviours if you had more information about the sustainability of the site? | 89.0 | 11.0 |
| 4. Do you think about why a site becomes a tourist destination? | 81.2 | 18.8 |
| 5. Do you consider your impact on the local environment and economy as a tourist? | 86.0 | 14.0 |
| 6. Would you be more inclined to visit a tourist site if you knew it was going to disappear in the coming years? | 78.8 | 21.2 |
| 7. Do you think your decisions as a tourism consumer would change if you knew more about sustainability? | 84.8 | 15.2 |
| 8. Do you think tourist sites should have sustainability management plans in place? | 97.0 | 3.0 |

### 3.1.1. Question 1—Do You Consider the Sustainability of Tourist Sites Prior to Visiting?

Question 1 asked respondents whether they considered the sustainability of a tourist site prior to visiting that site. Of the 165 respondents, 52.1 per cent indicated that they did not consider the sustainability of the site prior to visiting. The remaining 47.9 per cent indicated they did consider the sustainability of the site.

What this tells us is that, whilst a close result, the majority of consumers surveyed did not hold sustainability as a primary motivator. This shows that there is still a growing awareness of sustainability issues at tourist sites, and there is room for greater information sharing. Interestingly, attitudes and behaviours of individuals when on holiday do not always reflect their normalised behaviours in regular life [46]. Previous research found that members of an environmentalist group did not always behave in ways that aligned with their values when on holiday [48]. Of note, however, is that many behavioural studies have shown that pre-existing attitudes, whilst a strong indicator of intention, do not always translate to behaviours [40,46]. Attitudes can therefore be influenced by a range of mediating factors, and as such, the survey questions assess behavioural decision-making habits [46].

### 3.1.2. Question 2—Was Your Decision to Visit Tourist Sites in the Last Year Informed by the Environmental, Cultural, or Historical Significance of the Site?

Question 2 asked respondents whether their decision to visit a tourist site in 2019 was informed by the environmental, cultural, or historical significance of the site. The purpose of this question was to gauge the level of awareness of individuals in informing their decisions. Of the 164 respondents (one did not answer this question), 69.5 per cent indicated that their decision to visit was informed by the significance of the site.

Decision-making processes are influenced by a number of psychological and non-psychological factors, including the consumer's needs, access to alternatives, the influence of social groups, economic value and usefulness, and personal satisfaction, which is linked, in turn, to entertainment and competence [49]. Such decision-making processes can be reduced to three dimensions: the need for satisfaction, social agreement, and travelability [50]. Through this lens, the results of the survey tell us that there may be a growing self-awareness of the tourist decision-making process, or at least greater knowledge around the purpose for visiting a site. This aligns with the theory that the more the tourist is engaged with information and ideas of sustainable tourism, the more informed and responsible their behaviour becomes [49].

### 3.1.3. Question 3—Would You Adjust Your Behaviours If You Had More Information about the Sustainability of the Site?

Question 3 asked respondents whether they would adjust their behaviours if they had more information about the sustainability of the tourist site they were visiting. This question was designed to test whether tourist awareness to more responsible behaviours might influence a behavioural change. Of the 164 respondents (one did not answer this question), 89 per cent indicated they would alter their behaviours when provided with more information about sustainability.

The difficulty in changing behaviours in the tourism industry thus far is that the industry operates in a hedonic context [51]. As a consequence, tourists primarily see their output and do not immediately see benefits in adjusting their behaviours as they might at home [51]. The results of the survey show that there is a willingness for course-correction in tourist behaviours, and therefore suggests that tourists desire to be informed on sustainability management.

### 3.1.4. Question 4—Do You Think about Why a Site Becomes a Tourist Destination?

This question was designed to once again measure the awareness and thought process behind tourist decisions. Of the 165 respondents, the vast majority (81.2 per cent) indicated that they did think about why the site became a tourist destination. This is a notable increase from question 2 (69.5 per cent indicated yes), where it was examined whether decisions to visit tourist sites were informed by the environmental, cultural, or historical significance of the site. The combination of results from question 2 and question 4 show a variance between tourists understanding why a site is a tourist destination and their decision to visit that site. Whilst more than two thirds of respondents choose to visit a site for its importance, more than 80 per cent make an additional cognitive effort in understanding why sites become attractive tourist destinations. The results of question 4 show a promising engagement on the tourist's behalf with the site and understanding its importance. Shifts in responsible behaviour are often as a result of increased social engagement, which influences attitudes and can in turn influence intentions and behaviours [49]. Respondents, however, understand the purpose of a site, suggesting that their decision making is informed, and this should therefore be replicated in their understanding of why a site needs to be preserved for future enjoyment and engagement.

### 3.1.5. Question 5—Do You Consider Your Impact on the Local Environment and Economy as a Tourist?

Question 5 asked respondents whether they considered their impact on the local environment and economy as a tourist. The purpose of this question was to measure the breadth of consideration of the tourist, whether it was reserved for the site itself or extended to sustainability issues in the community. Of the 164 respondents (one did not answer), 86 per cent indicated that they did consider their impact on local environments and economies. Previous research shows that increased social engagement empirically incites the tourist to adjust their behavioural intentions towards sustainable tourism [49]. Whilst promising, intentions do not always translate to action or a shift in responsible behaviour, unless they are within the same context (i.e., preconceived intentions versus immediate intentions). This suggests the that tourism experiences at sites would benefit from extension into local communities, as there is a clear willingness and growing awareness of the impacts. Alternatively, it may present an opportunity to leverage the theory of vested interest, where the object of tourist attitudes is both important and hedonically relevant [46,52]. It is important to note that sustainability is about a measured approach to balancing issues of the environmental, economic, and socio-cultural, and that the impacts on the local economy may be positive and encourage growth [53].

### 3.1.6. Question 6—Would You Be More Inclined to Visit a Tourist Site If You Knew It Was Going to Disappear in the Coming Years?

Question 6 was designed to gauge the influence of last chance tourism on consumer decisions and whether this would encourage greater traffic through the site, potentially leading towards greater

damage or accelerated disappearance. Of the 165 respondents, 78.8 per cent indicated they would be more inclined to visit a site if they thought they had limited time left to see it. Last chance tourism is when tourists seek experiences with vanishing landscapes, natural sites, or heritage, on the premise of seeing something before it is too late, and whilst a niche market, may gain popularity [54]. The concept extends on the idea of not just seeing the site, but proving one's presence before the opportunity is gone [41]. Last chance tourism in this context refers to sites suffering from human impact, either immediate or systemic (i.e., climate change), and does not include sites that may suffer natural disasters, as, without warning, there would be no opportunity for tourism. The concept in itself is paradoxical: increased traffic may accelerate the disappearance. Relatedly, however, a study conducted at the Great Barrier Reef found that often those motivated by last chance tourism were also more concerned about the health of the reef [54]. Interestingly, last chance tourism could prove beneficial. In some cases, for example, this may generate necessary funding and publicity for the conservation of these locations, as witnessed in the Maldives, where mass tourism has provided revenue to respond to climate change [55]. The finding of this question is that consumers are still motivated by last chance tourism, and it may be up to industry players how to leverage this business.

### 3.1.7. Question 7—Do You Think Your Decisions as a Tourism Consumer Would Change If You Knew More about Sustainability?

Similar to question 3 and behaviours, question 7 assessed the willingness of people to change their decision-making process when provided with greater information. Of the 165 respondents, 84.8 per cent indicated that they think their decisions would change if they knew more about sustainability.

As previously discussed, social engagement is a key factor in influencing decisions, as well-engaged tourists are more likely to consider impacts in their decision [56]. What this result suggests is that the onus is on the industry to increase awareness and information-sharing to influence consumer decision-making. Marketing can play a role in influencing how information is consumed and introduce relevant sustainability conversations that then bridge the gap between translating attitudes into behaviours [57]. Of the respondents that answered yes to question 3, 92.5 per cent also answered yes for question 7, showing that a shift in behaviours might produce a commensurate shift in decision making.

### 3.1.8. Question 8—Do You Think Tourist Sites Should Have Sustainability Management Plans in Place?

Question 8 was included in the survey to compare consumer expectations with the current state of sustainability management in tourism. Of the 164 respondents (one did not answer), almost the totality (97 per cent) agreed that tourist sites should have sustainability management plans in place.

When considering conservation, it was found that 50 per cent of World Heritage sites do not have tourism management plans featuring sustainability [18]. That figure is disparate with the findings of question 8, where nearly all surveyed tourism consumers agreed that sustainability management plans should be in place for tourist sites. Whilst the managers of protected areas and sites are often under competing pressures to deliver economic stability, educational experiences, and environmental integrity, the conservation of the site should be their first and foremost priority to guarantee the continued existence of the site [58]. The result of this question also raises additional questions concerning who regulates management, whether it becomes compulsory, and what the role of the consumer is. Further, this result suggests that there is an opportunity to develop a form of participatory planning where tourists and consumers play a role, as studies have shown that a consumer's involvement in planning and decision making helps to alter their own habits.

### *3.2. Interview Analysis*

Six interviews were conducted with a range of field experts or representatives from tourism bodies. Interviews consisted of seven open-ended questions that invited a discourse on issues within the tourism industry pertaining to sustainability and why the industry operates as it does (Table 4).

**Table 4.** Interview guiding questions and interviewed experts/representatives. The pseudonyms A, B, C, D, E, and F were used to conceal the identity of the interviewees.

| Interview Guiding Questions | Interviewed Field Experts/Representatives |
|---|---|
| 1. What do you think are the biggest issues facing the tourism industry in regard to sustainability?<br><br>2. Do you think tourism could be considered a sustainable industry?<br><br>3. Why do you think many tourist sites do not have sustainability management plans in place?<br><br>4. Do you think sustainability is approached differently in regard to the type of tourist site? i.e., cultural, environmental, historical.<br><br>5. What balance should be struck between managing the sustainability of the tourist site versus the sustainability of adjacent issues such as local economies and cultural integrity?<br><br>6. What could be done to influence the tourism industries' response to sustainability?<br><br>7. How much weight do human behaviours carry in regard to sustainability and how can we best influence this? | • Senior lecturer in Tourism Programs at a University in Western Australia (A)<br>• Park Manager at a National Park in Australian Capital Territory (B)<br>• Manager of a National Park in New Zealand (C)<br>• Tourism Sustainability Advocate in a New Zealand Tourism Industry Authority (D)<br>• Head of Sustainability in a Tourism Authority in South Australia (E)<br>• Tourism Officer in the Northern Territory (F) |

The scope of experience and opinions from these experts and tourism representatives provides a picture of the current state of sustainability in the tourism industry. Collectively, they agreed that isolating one definition of sustainability in tourism was difficult, as it was dependent on the perspective and end goals of sustainability. They also agreed that tourism may look different in the aftermath of the coronavirus pandemic. In the analysis below, experts and tourism representatives will be referred to by their last name.

### 3.2.1. Question 1—What Do You Think Are the Biggest Issues Facing the Tourism Industry in Regard to Sustainability?

Whilst a broad question, discussions with the interviewees found some consistent themes of issues within the industry, and F proposed that sustainability is viewed differently by industry players (hotel chains to small operators). F identified that, within the Northern Territory, a continued focus is on the notion of offsetting how tourists get from point A to point B, as transport (car and air travel) remains an issue. B proposed that one of the key issues for the industry was managing people, as nature manages itself, and that so many National Park projects (e.g., infrastructure, camp sites) are for the purpose of lessening the impact of people on the environment. Along these lines, a discussed a gap in aligning how changes in policy and management of sustainability will improve the tourist experience. Problematically, sustainability as a concept may be off-putting to the tourist if they believe it to be inaccessible, and there is an onus on tourism providers to communicate how shifts in personal action will benefit the tourist. For example, during a turtle watch, tourists who followed instructions to not shine their torches had more rewarding experiences and saw more turtles. B similarly said that there is a greater need for individuals to see how they interact with the landscape and to view themselves as part of the environment. This idea has been previously captured as place attachment and suggests that visitors are more inclined to make responsible decisions when they feel a connection to the site [40].

C, D, and E discussed the increased movement and volume of people as a fundamental issue. D outlined how marketing has encouraged increasing numbers of tourists to visit countries without a proper focus on managing these people, globally. Similarly, E identified that concentrated tourist numbers at peak times caused the most impact but supposed that adjusting the flow and rhythm of visitors to sites may soften these impacts and bolster the experience (shifting to a value proposition approach). The interviewee E discussed how growing tourist numbers often dominates conversations,

but that growth and counting tourist numbers is no longer an acceptable metric in measuring the welfare and wellbeing of a site. Future measures should assess whether the tourism experience has detracted or added to the future enjoyment of the site.

### 3.2.2. Question 2—Do You Think Tourism Could Be Considered a Sustainable Industry?

The discussion around the idea of sustainability as a concept has defined the answers to this question, as it depends on the lens under which it is viewed. Interviewee A, for example, captured the deviations in definition in showing that sustainability for an environmentalist means an untouched landscape, whereas banks and government view sustainability in terms of growth, and as such, it is overwhelmingly about balance. Additionally, E agreed that there needs to be a balance against the impacts of people, planet, and place, and that some areas of the industry are sustainable, but that often those that are not are symptomatic of globalisation. On the other hand, D argued that there is a long way to go in balancing sustainability efforts and then moving towards regenerative efforts, both environmental and community. D also indicated that any changes need to be done in consultation and partnership with local communities, and that for the industry to be sustainable, there needs to be a significant reduction in carbon footprint and air travel.

The interviewee B suggested that the industry could become sustainable if the tourist develops a broader understanding of how they fit into the environment and uses that as a starting point for better viewing themselves as a mechanism of the industry. C also agreed that the industry could be sustainable, but that it would require game-changing thinking that not only addresses flow, volume, and movement but also the macro issues of climate change and fuel/energy consumption. The industry leans towards incremental shifts in approach that do not see substantial change. F argued that the industry is sustainable in the scope of the Northern Territory and for nature-based sites, as the tourism experience is founded on the preservation and longevity of that site. This is consistent with theories within sustainability leadership, where there is a growing awareness that their success is dependent upon the context and environment in which they operate [19].

### 3.2.3. Question 3—Why Do You Think Many Tourist Sites Do Not Have Sustainability Management Plans in Place?

Sustainability as a concept continues to shift in definition and scope. In this context, E suggests that sustainability has become an unwieldly concept for tourism operators to approach, as one size does not fit all. The vast majority of tourism businesses in Australia are small operators, and they do not have the time, expertise, or funding to address sustainability in their business. D continues that people were slow to get involved with sustainability, possibly occupied with the surprising growth in tourism and financial gains. A succinctly captures the issues as cost and expertise. There needs to be greater focus on convincing people and tourism providers of the value in it for them, whether that is economic, future, personal, or existence value. C also suggests that the tourism and economic systems provide no incentive for businesses to change their behaviours, a and that an extra cost for small operators would not be economically sustainable for their business model. Similarly, whilst F indicated that for the Northern Territory, most large blocks of land (tourist assets) are under protected area management obligations, he suggested that a limited access to funding and differences in governance approach may result in fewer management plans. B indicates that consumers view their impact within their perspective of time (their life span), wheres the environment exists on a time scale of hundreds of thousands of years. People do not see the long-term impacts of their current actions, and therefore operators also have short term views.

D suggested that in lieu of official management plans, sites could charge the true cost of travel (including carbon footprints), resulting in fewer tourists, or introduce tourist limits similar to those at Machu Picchu and the Galapagos Islands that allow for better management.

3.2.4. Question 4—Do You Think Sustainability Is Approached Differently in Regard to the Type of Tourist Site (i.e., Cultural, Environmental, Historical)?

The approaches to sustainability are less dependent on the type of tourist site it is, but rather as a result of the context in which the site exists. More often than not, the approach is informed by the values of the area and the guardians of the site. B indicated that in the context of national parks; cultural, environmental, heritage, and biodiversity values are interlaced, and the success of these values are as a result of them being shared by the custodians, operators, and visitors. Similarly, a referred to sustainability theoretically as a balance of these pillars, where any approach should integrate all at once. F again discussed the idea that if the tourist site does not exist, regardless of its type, there is no product to encourage tourism. In the context of the Northern Territory, in Australia, sites and national parks are jointly managed with traditional owners and those most connected to the land. Strong leadership and joint management are a crucial pillar in maintaining positive development of all the pillars of sustainability within the tourism industry. D agreed that the varying levels of management of sites within the respective categories is dependent on how connected people are to the site and how passionately they feel about preserving it. Using the Fiordland National Park as an example, C discussed that many of the approaches are cognisant of what is needed to operate, and that many of the values in New Zealand already align with environmental, community, and business sustainability. E suggested that the idea of a sustainable business is that bringing people to connect with a site leaves a positive impact on the visitors and the site, but in the same vein, profit is essential to operate and drive the business.

For New Zealand, preserving the Maori heritage is increasingly becoming a sustainability priority, with *kaitiakitanga* (Maori term for guardianship and conservation) being woven into experiences, destinations, and sites as an important pillar of conservation and restoration. This was echoed by F in the management of national parks in the Northern Territory. Whilst studies have observed the socio-economic benefits of indigenous tourism, achieving a balance between growth and cultural sustainability is reiterated [59].

3.2.5. Question 5—What Balance Should Be Struck between Managing the Sustainability of the Tourist Site Versus the Sustainability of Adjacent Issues such as Local Economies and Cultural Integrity?

The sustainability of the site itself and the sustainability of adjacent industries and issues should not be mutually exclusive. C, D, and E all agreed that these sustainability priorities need to be integrated and interlinked in management. D discussed how these adjacent sustainability issues are sometimes managed in isolation, when planning should be done in coordination with the community. B discussed that the balance should be struck based on an appreciation for the long term, with a lens of longevity and adopting an intergenerational aspect. F suggested that without an economic return from the site to the community or the government, the conservation value cannot always be warranted or explained. Interviewee A explained the nexus that if the local environment is not protected to be attractive, the economy will suffer in tandem.

"If you destroy the asset on which your economy is based, then the economy will fail as well. Planning for economic sustainability needs to internalise the costs and values of that natural environment"—A, 2020.

Additionally, a also discussed the usefulness of environmental economics in this instance and the fact that once an environmental issue is converted to a monetary value, people can see the cost to benefit ratio. People also need to understand the practicality behind management rhetoric, and this should be conveyed through clear policies. E socialised the idea of economic nutrition using the example of Luxury Lodges Australia (19 lodges in 17 regions), who partner with over 1600 small and regional businesses that become visible to their guests and create products and experiences of the place. Evidence from these findings demonstrates that none of these pillars of sustainability can be managed in isolation, tourism impacts and is in turn affected by any major changes within these concepts, and long-term planning needs to balance the longevity of each [60].

### 3.2.6. Question 6—What Could Be Done to Influence the Tourism Industries' Response to Sustainability?

Joint partnerships, awareness ventures, and specific leadership are the main potential industry and government responses to sustainability. F, A, and C expressed an interest in the role of government in influencing the industrial response. F indicated there may be room to replicate the governance that occurs in Protected Area Management across other aspects of the industry, but for that to work, it would need to not be overdone and work closely with indigenous leaders and tourism providers. Additionally, C specified that a more cohesive approach from central government was needed. D suggested that targeted investment from government into railways, hydro-powered transport, and other carbon-neutral options may assist the industry. In the New Zealand context, there is room to link finance agencies with tourism and conservation efforts and improve the way tourism is interwoven into the fabric of the way government operates (in comparison to farming, which is supported by agencies and even rural bankers who understand their business). This idea of the industry not being understood was replicated by F, who suggested that whilst government grants may help the industry reshape their models towards sustainability, financially their balance sheets would reflect a loss.

In this context, a discussed the idea of local people taking up leadership roles in communicating the community's values, an approach reminiscent of the idea of the social change model of leadership, which encourages behavioural changes and individual agency [61]. A also discussed that industry and government approaches need to be careful of the rhetoric and imposing specific values on users who have differing values without explaining the ideas behind them. E indicated that there was a flood of good sustainability intentions, but ensuring the industry converts these intentions into behaviours remains a goal, and giving visitors agency and explaining the rhetoric behind sustainability may support this.

B suggested that the industry needs broader awareness programs through a conservation lens that provide operators with necessary information so that they can be the conduit for information flow to the consumer. Interestingly, findings from Q3 of the survey suggest that consumers would change their behaviours at tourist sites if provided with more information and thus supports this recommendation. Once informed, providers and industry operators can also engage with differing parts of the community about conservation values and the benefits of sites (such as the Namadgi National Park, which provides 80 per cent of Canberra's water sources) to create advocacy.

### 3.2.7. Question 7—How Much Weight Do Human Behaviours Carry in Regard to Sustainability and How Can We Best Influence This?

Human behaviours are at the crux of sustainability issues, as without human interaction, natural environments do not need to be managed. When on vacation, humans typically revert to the easiest option, reflective of the hedonic nature of tourism, and as creatures of routine, it is hard to change behavioural traits and will likely require incentives. To be able to influence the behaviours of people as visitors to a site, their behaviours first need to be understood.

Interviewee E discussed Ross Honeywill's economic theories around the new economic order and more specifically, the concept of the desire economy, where people select things that align with their values, rather than being dictated by cost (the concept of planet traditional) [62]. Further, E suggested that operators should advertise what their value proposition is, ensuring it aligns to the values of their clients and how it connects to planet, people, and place.

A suggested that people are more receptive to messages that align the importance of the issue to the human experience, using the 2019/20 Australian Bushfires as an example of the need for increased tourism in those communities hit by the fires. Thus, a narrative needs to be developed around the importance of sites and issues that appeal to the human condition, and as B suggested, conveying the message that we are only temporary custodians of the lands.

For New Zealand, C suggested that governments are too risk adverse to introduce major changes due to short election cycles, and thus incremental changes that influence human behaviours could

be introduced, such as reusable water bottles or soap bars with no packaging in hotels. D confirmed that these incremental changes need to be made easy for tourists, so they encourage responsible decision-making. For example, the 'Tiaki Promise' in New Zealand encourages visitors to care for the land, but does not show them how to do that, which would be the next stage. D also indicated that there is a role for leadership to communicate how things are done in the community and to lead by example, again reflecting the idea of social change leadership.

One of the difficulties in shifting existing behaviours is the contested definition of sustainability, as it means different things to different people, industries, and governments. There is no rationale for a global definition of sustainability, and it is largely informed by local settings, as it should be. A suggests that perhaps sites or tourism bodies could develop a framework that helps to inform, rather than dictate, responsible decision making. Despite this, F believes that as more people and future generations become aware of a mainstream definition of sustainability, it will become part of the generational knowledge, and behaviours will shift over time to replicate what is considered socially acceptable.

## 4. Discussion—Management Strategies for Sustainable Tourism

Considering the combined quantitative (survey) and qualitative (interviews) research, four managerial strategies towards sustainable tourism sites have been identified. The results of surveys and interviews reveal that tourists typically have positive sustainability intentions, which, however, are likely not being converted into behaviours and attitudes when participating in the tourism experience. The following proposed strategies were planned to bridge the gap between the discovered intentions and converting them to practice.

### 4.1. Cost Strategy: Pricing Tourism Experiences According to a Value Proposition

The interviews suggest that one of the biggest issues facing the tourism industry is managing the movement, volume, and behaviours of people. Interviewees suggest that human behaviours are at the crux of tourism sustainability issues, as their behaviours, consumption, and mass movements provide pressure on natural environments. Interestingly, survey results seem to indicate that there is consumer goodwill and a willingness to participate in sustainability efforts. Indeed, a majority of survey participants indicate that they would change their behaviour (89 per cent) and decisions (84.80 per cent) if provided with more information about sustainability.

Interviewed expert opinions agree that sustainability will always depend on economic balance, and that individual sustainability pillars (economic, socio-cultural, or environmental) cannot be favoured over another. As a consequence, one pillar will falter if another does (i.e., no economic gain if the site is diminished). The first recommended strategy proposes to increase the cost of tourist experiences to match the value proposition of the site and associated business (Figure 2). Revenue made could be reinvested into conservation and would help to offset the impacts of tourism. The idea also leans towards the concept of the desire economy, where people will pay for an experience that aligns with their values [62]. Critical to the effectiveness of this strategy is for people to understand the shared values and experience they are receiving [63]. Converting the damages and impacts on the site into a monetary value helps it to be understood.

This strategy may also be an effective tool in managing the increase in last chance tourism. Indeed, 78.8 per cent of survey respondents indicated that a disappearing site would act as a motivator to visit, and whilst the collected revenue helps to conserve the site and bring awareness, increased numbers may strain sites like the Great Barrier Reef further [54,64]. This is where a cost strategy may be most effective in collecting revenue, creating awareness, and drawing visitors who share the values of the site.

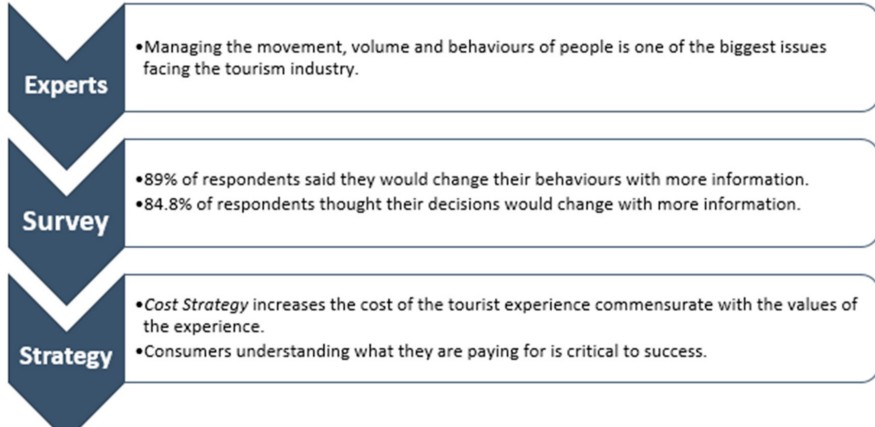

**Figure 2.** Cost strategy informed by observations from interviewed experts and surveyed individuals.

*4.2. Framework Strategy: Guiding Businesses to Operate Sustainably*

Survey respondents almost unanimously supported (97 per cent) the idea of tourist sites and businesses having sustainability management plans. In discussions with field experts and representatives, they indicated that most sites may not have sustainability plans in place due to a confusing scope of sustainability as a concept, the cost involved, and a lack of expertise. What this suggests is that if it were made easier and more cost effective for businesses to introduce such plans, it would be met with support from their consumers.

The second recommended strategy is a government-led framework designed in partnership with industry, which advises businesses how to act sustainably and discusses the concept with their clients (Figure 3). Whilst a framework might not fully capture the many elements of sustainability and may not target all business types, it would reduce the cost and lack of expertise for smaller businesses in developing their own plans. The purpose of such a framework is to make sustainable business decisions and behaviour easier for people who are time poor, or do not have the resources to introduce something similar themselves. The framework could also provide a definition of sustainability to guide businesses and acknowledge that continuous growth in the face of a diminishing site is not sustainable in the long term.

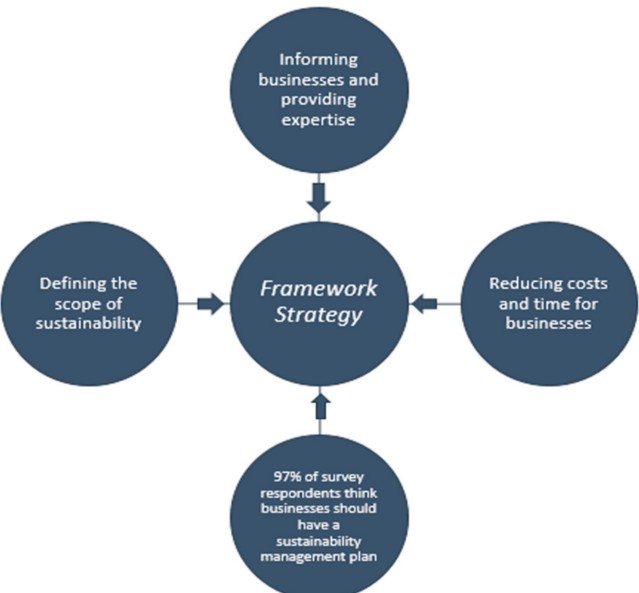

**Figure 3.** Capture of the framework strategy, showing the key features the strategy will address.

The role of government in tourism and destination management has increased since the G20 meeting recognised tourism as an avenue for growth and employment, however it has not extended to sustainability [65]. Many sustainability frameworks were initiated at a grassroots level by universities or consultants, such as the Cuban Tourism Industry and a framework for sustainable tourism in Aruba, and this suggests that governments have not previously delivered such a product [66,67]. Additionally, variances between local and national approaches allow for the dominance of private interests, rather than strategies focused on equitable and sustainable development [68].

*4.3. Marketing Strategy: Advertising the Importance of Sustainability to the Tourist Experience*

Interviewed experts suggested that there needs to be a narrative around the human relationship with tourist sites and ancillary industries. The goal of this relationship is to appeal to the human condition, align sustainability efforts to the visitor experience, and help people to understand their impact on tourist sites. This idea of understanding the purpose and importance of tourism is supported by the survey results. Indeed, 81.2 per cent of participants indicated they consider why a site becomes a tourist destination and 86 per cent indicated they already consider their impact on the environment and economy as a tourist. Such results suggest there is a foundational thought process in consumers that could be developed further, and survey results suggest this educational piece could be started earlier in the tourism experience (only 47.9 per cent of respondents indicated they think about the sustainability of a site prior to visiting).

The recommended third strategy is for businesses to better integrate issues of sustainability into their marketing and advertising, leveraging their built knowledge and plans from the framework strategy (guiding businesses to operate sustainably) to better communicate the importance of sustainability (Figure 4). Discussions with field experts show that shifts in visitor behaviour are in the interest of the business, as they will preserve the tourist site, and thus the basis for their business and economic return, for a longer term. A study on Chinese visitors in New Zealand found that more visitors were aware of the idea of sustainability as part of a tour group (59.6%) than independent travellers (40.4%), suggesting that tourism operators and businesses have an impact on the individual in how they run and communicate their business [69].

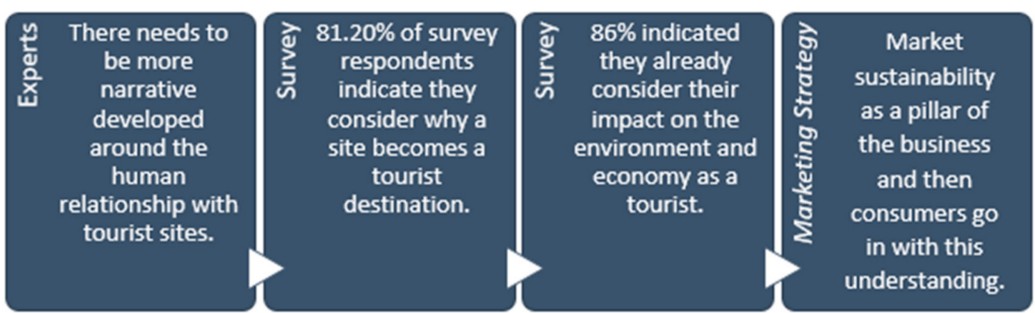

**Figure 4.** Marketing strategy: observations from interviewed experts and surveyed individuals that inform the strategy.

Marketing has been an effective tool in positioning profit-targets and demand, and thus can play a role in shifting visitor behaviours. Locations like English National Parks (UK) have focused on socio-economic marketing, and sustainability has been marketed by environmental organisations [70]. This altered form of marketing could be classified as social marketing, where commercial marketing may be introduced to induce behavioural change [71]. Tourism marketing has a system-wide effect, and rather than focus on arrivals, this could focus on the quality of experience and sustainability [72]. Marketing sustainability and focusing on the quality of the experience will also provide businesses adopting the cost strategy (pricing according to value proposition) with greater justification.

Marketing and advertising continue to shift online to services that enable consumers to engage with businesses, and as such will only be a minimal maintenance cost for small businesses. This will not only benefit the sustainability of the tourist site and their business, but will leverage the behavioural willingness found in the survey results. Consumers will, however, need to be savvy in determining whether a business is simply using sustainability marketing to appear more ethical, rather than having an actual commitment to the premise [73].

*4.4. Partnering Strategy: Working with Local Custodians to Enhance Value*

The interviews with field experts in New Zealand and the Northern Territory (Australia) highlight the importance of partnering with indigenous people and local custodians who have an enduring understanding of the land and needs of the site. This is a two-way dynamic, as not only do increased knowledge and values inform decision making, but they allow for indigenous people to take ownership of their tourist sites [74]. 69.5 per cent of survey respondents indicated their decision to visit a site was dictated by that sites' importance and significance, showing that there is an opportunity to build awareness of the importance of certain sites of cultural value. For example, many New Zealand tourism policies are infused with the Maori principles of guardianship and care for the land. Integrating principles from indigenous peoples within site management is therefore highly beneficial, and could be better replicated in many other countries, such as Australia, Greenland, or Scandinavian countries.

The fourth recommended strategy proposes to increase partnerships with local custodians to integrate principles of respect, guardianship, and care into tourism experiences and policies (Figure 5). Interviewees suggested that people are more inclined to make responsible decisions when they understand the purpose of a site and it appeals to the human condition. Studies in Carey Island (Malaysia) show that personal connections and indigenous knowledge were key to developing tourism [75]. This strategy would also leverage the noted history of inherent sustainability practices in indigenous cultures and provide Australian tourism with a point of differentiation, notably an increased interest in indigenous tourism [76]. This integration of principles of respect and guardianship into the visitor experience will also make it more authentic and thus strengthen the value proposition of the cost strategy.

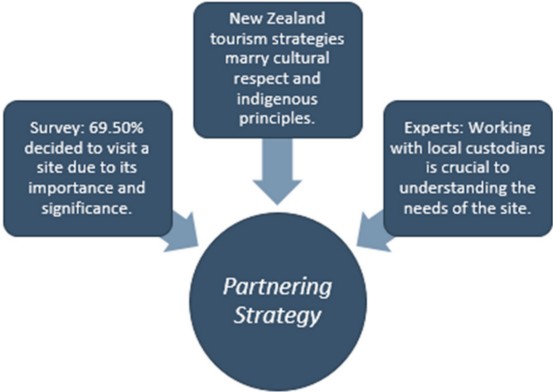

**Figure 5.** Capture of the partnering strategy, including observations from interviewed experts and survey results that inform the strategy.

Additionally, this strategy would include the local communities as the custodians of tourist sites. In a study in Zimbabwe, it was found that, despite being custodians, local people were not always included in decisions about their own heritage or cultural destinations, nor did they reap the rewards of expanded tourism [77]. This was echoed in Tanzania, where the inability for local custodians to manage their own resources and contribute to planning was a detriment to heritage site management and thus bred systemic apathy towards conservation efforts of heritage and cultural tourist resources [78].

What this tells us is that partnering with local communities, custodians, and indigenous leaders is in the interest of all parties and the industry.

## 5. Conclusions

Studies and conducted research empirically show that there is a willingness on the behalf of the visitor to be better informed on issues of sustainability as it pertains to tourist sites, and that this information may influence their behaviours and decision making. Quantitative survey results show that a majority of survey participants who travelled in 2019 were either already conscious of their impact on the local community (environment and economy) or were willing to build an increased awareness on the topic. Discussions with field experts also indicated that a major hurdle in addressing sustainability in tourism was aligning the interests of the site and environment to the individual visiting and building that awareness in the visitor that whilst their movements were transient, their impact was permanent. Results from the survey and interviews conducted in this research make significant practical contributions, outlined in more detail in the following sub-section.

Literature previously framed the changing of social behaviours and choices of individual actors as a critical element to achieving sustainability. Whilst this conversation was framed through the lens of marketing [57], this study suggests that there needs to be more accountability on the industry and tourism bodies to influence these behaviours. The onus cannot completely be upon the individual to shift towards what is considered responsible decision making without it being made easy for them or without building the awareness of how their behaviours impact the longevity of a site.

Previous literature suggests one of the primary problems within tourism sustainability has been the fragmented and inconsistent response to conservation and site management. Whilst this research suggests that there are opportunities for greater focus on site management and integration of sustainability mechanisms in planning activities, achieving a consistency in site management is less of a priority than shifting the behaviours and attitudes of those that visit it. Management approaches should be site-specific and contextual to the environment, where behavioural shifts can be addressed more collectively across the industry.

The main contributions of this research are the proposition of four managerial strategies that can be adopted by governments, tourism bodies, and business managers to their individual circumstances, to manage tourism sites and businesses most effectively towards environmental, social, and economic sustainability. A cost strategy recommends pricing the visitor experience against the actual cost (both immediate and future expenses, including preservation) and by presenting this cost as a value proposition. Critical to the cost strategy is informing the visitor on the values they are paying for. A framework strategy recommends a government-led framework that provides tourism businesses with the supporting information and guidelines by which to operate sustainably and should consider the critical balance of site preservation to economic return. A marketing strategy recommends tourism businesses advertise the issues of sustainability and conservation in their materials and services to integrate the conversation and awareness from the beginning of the business–client relationship. Lastly, a partnership strategy recommends increased cooperation with local communities and custodians in the delivery of tourism services so that visitors can witness a connection with their experience and the values of the people that connect with the site.

Findings from this research suggest that residual goodwill and willingness on the behalf of the visitor to behave more sustainably is not being translated to actual behaviours at tourist sites. There is disconnect between human behaviours in their ordinary habitat (home) versus when they are visiting, and this is synonymous with the idea of the holiday, which presents an escape from ordinary behaviours and values [46]. Future research will need to be done on assessing how the survey results would translate into the actual tourist environment. The results of this paper show the willingness on behalf of the visitor and the industry to improve on sustainable behaviour and responsible decision making; however, observing these intentions in practice was outside the research boundaries and

further study on this will help understanding of where the pressure points in successfully changing behaviours lie.

The scope of this study was impacted by two main limitations. First, as the survey was conducted and advertised through social media, the target market of the respondent may have leaned towards certain demographics. This limitation was addressed by expanding the demographic and targeting additional age demographics and groups. Further, as part of the analysis of the survey relies on theories surrounding comparing human attitudes to behaviours, and that there is sometimes a deviance between these, it is pertinent to note that without a practical assessment of the same respondents, we cannot know if the opinions they expressed in the survey would translate to behaviours. There is also the factor that respondents will have completed the survey in their home environment surrounded by ongoing practices, which again may not translate in a tourist environment due to the hedonistic nature of 'vacationing'.

Second, the impact of the COVID-19 pandemic must also be accounted for in the limitations, as it has severely impacted the tourism industry and individuals' capacity to travel. As the pandemic initially prevented international travel and then domestic, it became impossible to conduct surveys and case studies in person at tourist sites and that shifted the methodology. In addition, whilst survey participants were completing the survey regarding travels conducted in 2019, it must be assumed that they would be viewing some of the questions and answers through the lens of the current pandemic, which may have influenced their thinking.

**Author Contributions:** Conceptualization: R.M.M. and R.M.; methodology: R.M.M.; formal analysis: R.M.M.; investigation: R.M.M.; writing—original draft preparation: R.M.M.; writing—review and editing: R.M.M., R.M. and G.M.M.; supervision: R.M. and G.M.M. All authors have read and agreed to the published version of the manuscript.

**Funding:** This research received no external funding.

**Acknowledgments:** The authors wish to thank J Carty for additional editing and review.

**Conflicts of Interest:** The authors declare no conflict of interest.

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
