# Peer review of "Strategic Decisions for Sustainable Management at Significant Tourist Sites"

_sustainability, doi:10.3390/su12218988_

Round 1
Reviewer 1 Report
The contribution is correct from the point of view of the methodological approach: the part of the theoretical framework is good and useful; the empirical investigation section appears interesting and original. The survey is conducted carefully and takes into account a number of variables.
The bibliographic references are sufficient considering that a relevant part of the contribution is the original work of the authors.
The clarity of presentation is adequate. Overall, the work needs to be reread and requires more attention to the proposed form.
The contribution is original and innovative. It is therefore considered, as the final judgment of the referee, that the contribution can be accepted.
Reviewer 2 Report
It should be better, in the whole document, to write in passive. For example instead of "In this research we study" - "In this research it has been studied").
Line 19 instead of mixed methodology - combined methodology
Line 95 instead of ...we propose - it has been proposed
Line 97 instead of "To inform our strategies, we conducted both...." - "To inform proposed strategies, both quantitative studies......, have been conducted.
Line 117 insted of "In this paper we explore..." - In this paper it has been explored...
The first person plural should be replaced with the passive in the whole document.
Line 532 - Figure 2 is missing
In the conclusion the main contribution of this paper and the main limitation are missing.
Line 656 instead of our research - this research
Line 661 instead of our findings - The results of this paper show...
Reviewer 3 Report
I suggest to strengthen the introduction paragraph. The introduction should contain the main motivations of the study, the paper aim, the methodology and the structure of the paper.
I suggest to describe the objective in a clear way, in order to understand the theoretical gap that is intended to be filled. Moreover, it is very important, at the end of the introduction, to insert the structure of the work with a very small synthesis of the paragraphs. For example, the paper contains N ...paragraphs... Paragraph 1 concerns..... Paragraph 2 describes... ecc.
The main problem of this paper is that "Theoretical background" is missing.
I recommend adding the theoretical framework in which the research paper is placed.
Round 2
Reviewer 2 Report
No more comments
Author Response
We thank Reviewer 2 for the many valuable comments. These comments helped us to improve our manuscript.
Reviewer 3 Report
It is not sufficient to include theoretical elements in the introduction.
I recommend, again, the inclusion of a specific paragraph on "Theoretical background" and justifying the empirical research carried out.
I also suggest to delete the sub-paragraphs in the conclusions.
Author Response
Point 2.1: It is not sufficient to include theoretical elements in the introduction.
I recommend, again, the inclusion of a specific paragraph on "Theoretical background" and justifying the empirical research carried out.
Response 2.1: Thank you for this suggestion. We now include, in a separated section (Section 1.1) a more thorough justification of the research and its research design, L130-137.
Point 2.2: I also suggest to delete the sub-paragraphs in the conclusions.
Response 2.2: We have deleted the sub-paragraphs as suggested.
Round 3
Reviewer 3 Report
cari Autori,
The paper is very interesting, but not yet publishable. There is confusion between theoretical background and research design.
The theoretical background contains all the scientific references in the literature about "Strategic decisions for sustainable management", based on which the questions included in the survey and interviews have been identified.
The one currently present is the research design. The research design contains all the phases useful for the realization of the project and should be included in the Materials and Methods section.
I therefore recommend to write a theoretical background in depth and allow the reader to clearly identify the research gap and the drivers used in the empirical research phase.
Author Response
Dear Reviewer, thank you for this additional valuable comment. We do acknowledge that what we wrote in round 2 was our research design and have moved it as an introduction to the Materials and Methods section. Regarding the theoretical background, we now explain which observations have informed focus of this paper (L130-167).